# Selective Preference Aggregation

## Abstract

Many applications in machine learning and decision making rely on procedures to aggregate human preferences. In such tasks, individuals express ordinal preferences over a set of items by voting, rating, or comparing them. We then aggregate these data into a ranking that reveals their collective preferences. Standard methods for preference aggregation are designed to return rankings that arbitrate conflicting preferences between individuals. In this work, we introduce a paradigm for *selective aggregation* where we abstain from comparison rather than arbitrate dissent. We summarize collective preferences as a *selective ranking* – i.e., a partial order that reflects all collective preferences where at least $100 \cdot (1 - \tau)\%$ of individuals agree. We develop algorithms to build selective rankings that achieve all possible trade-offs between comparability and disagreement, and derive formal guarantees on their recovery and robustness. We conduct an extensive set of experiments on real-world datasets to benchmark our approach and demonstrate its functionality. Selective rankings improve reliability under distribution shift and adversarial manipulation by exposing disagreement and abstaining on disputed pairs.

## 1 Introduction

Many of our most important systems rely on procedures where we elicit and aggregate human preferences. In such systems, we ask a group of individuals to express their preferences over a set of items through votes, ratings, or pairwise comparisons. We then use these data to order items in a way that represents their collective preferences as a group. Over the past century, we have applied this pattern to reap transformative benefits from collective intelligence in elections [1], online search [2], and model alignment [3].

Standard methods for preference aggregation represent collective preferences as a *ranking* – i.e., a total order over $n$ items where we can infer the collective preference between items by comparing their positions. Real-world preference data are noisy, strategic, and shift across populations, making total orders brittle. Rankings reflect an *approximate* summary of collective preferences because it is impossible to define a coherent order when individuals disagree. This impossibility – which is enshrined in foundational results such as Condorcet's Paradox [1] and Arrow's Impossibility Theorem [4] – has cast preference aggregation as an exercise in *arbitration*.

Over the past few decades, we have developed countless algorithms from this perspective [see 5, 6] to reap benefits from collective intelligence in new use cases:

- Support Group Decisions – e.g., to fund grant proposals or hire employees [7, 8].

- Qualitative Benchmarks – e.g., to rank colleges [9], products [10], or language models [11].

- Model Alignment – e.g., to fit or fine-tune models whose predictions align with the preferences of their users [3, 12].

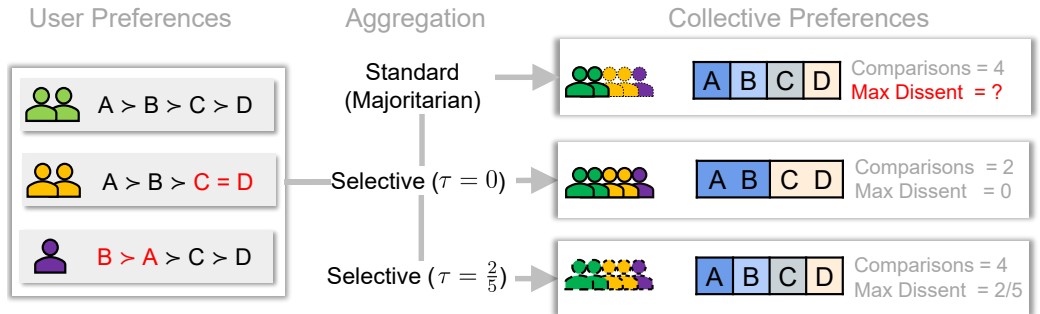

Figure 1: Comparison of collective preferences for $5$ users over $n = 4$ items. Standard rankings arbitrate disagreement and hide it. Selective aggregation returns a partial order (tiers): items in different tiers are comparable, and any such comparison overrules at most $100 \cdot \tau\%$ of users. The tiers make disagreement explicit — e.g., $\tau = 0$ gives unanimous $\{A, B\} \succ \{C, D\}$, while $\tau = 2/5$ recovers a total order if one accepts overruling up to $40\%$.

In many of these use cases, *we do not need a total order.* Abstaining on contested pairs and keeping only well-supported comparisons yields more robust outcomes. When we aggregate preferences to rank colleges, a total order can strongly influence where students apply and how institutions invest [see e.g., 13–16]. When we aggregate preferences to predict helpfulness [17], a total order can lead us to build models that are aligned with the preferences of a slim majority [12].

In this work, we propose to address these challenges through *selective aggregation*. In this paradigm, we express collective preferences as a *tiered ranking* – i.e., a partial order where we are only allowed to compare items in different tiers. We view tiers as a simple solution to avoid the impossibility of arbitration: given a pair of items where individuals express conflicting preferences, we can place them in the same tier to abstain from comparison. We capitalize on this structure to develop a new representation for collective preferences that can reveal disagreement, and new algorithms that can allow us to control it.

Our main contributions include:

1. We introduce a paradigm for preference aggregation where we summarize collective preferences as a *selective ranking* – i.e., a partial order where each comparison aligns with the preferences of at least $100(1 - \tau)\%$ of users.

2. We develop algorithms to construct all possible selective rankings for a preference aggregation task. Our algorithms are fast, easy to implement, and behave in ways that are safe and predictable.

3. We conduct a comprehensive empirical study of preference aggregation in modern use cases with diverse preference data. Our results show how selective rankings can promote transparency and robustness compared to existing approaches.

4. We demonstrate how selective aggregation can be used to learn from subjective annotations in a case study in toxicity detection. Our results show how selective aggregation can improve model performance and align predictions with a plurality of users.

5. We provide an open-source Python library for selective preference aggregation, available on anonymized repository.

**Related Work**

Our work is motivated by a growing set of applications where we aggregate conflicting preferences. In machine learning, such issues arise in tasks such as data annotation [18–20] and alignment [3, 21, 22] as a result of ambiguity, subjectivity, or lack of expertise [21, 23, 24]. In medicine, for example, conflicting annotations reflect uncertainty regarding ground truth [see e.g., 25–28]. In content moderation, conflicting annotations reflect differences in opinion [29, 30].

Our work is related to an extensive stream of research in social choice [31]. This body of work establishes the mathematical foundations for preference aggregation by defining salient voting rules and characterizing their properties [see 32, 33, for a list]. Although much of this effort is driven by the impossibility of reconciling individual preferences [see e.g., 4, 34], few works mention that we could abstain from arbitration by representing collective preferences as a partial order. Abstention is not a viable option in many of the applications that have motivated work in this field. In elections, for example, we cannot aggregate ballots into a partial order because we must identify a single winner [35].

On a technical front, our work complements a stream of research on rank aggregation [2, 36–38]. Although most work focuses on representing collective preferences as rankings, some focus on coarser representations such as bucket orderings [see e.g. 39–41, and references therein]. For example, Achab et al. [39] view bucket orderings as a "low-dimensional" total order and characterize their potential for recovery. Andrieu et al. [40] use them as a vehicle to efficiently combine multiple rankings. In general, these differences in motivation lead to differences in algorithm design and interpretation. For example, items that we would consider "equivalent" in a bucket ordering would be "incomparable" in a tiered ranking.

## 2   Framework

We consider a standard preference aggregation task where we wish to order $n$ items in a way that reflects the collective preferences of $p$ users. We start with a dataset where each instance $\pi_{i,j}^k$ represents the pairwise preference of a user $k \in [p] := \{1, \ldots, p\}$ between a pair of items $i, j \in [n]$:

$$
\pi_{i,j}^k = \begin{cases}
1 & \text{if user } k \text{ strictly prefers } i \text{ to } j \Leftrightarrow i \overset{k}{\succ} j \\
0 & \text{if user } k \text{ is indifferent} \qquad \Leftrightarrow i \overset{k}{\sim} j \\
-1 & \text{if user } k \text{ strictly prefers } j \text{ to } i \Leftrightarrow i \overset{k}{\prec} j
\end{cases}
$$

Pairwise preferences can represent a wide range of ordinal preferences, including labels, ratings, and rankings. In practice, we can convert all of these formats to pairwise preferences as described in Appendix A.2. In doing so, we can avoid restrictive assumptions on elicitation. For example, users can state that items are equivalent by setting $\pi_{i,j}^k = 0$, or express preferences that are intransitive. In what follows, we assume that datasets contain all pairwise preferences from all users for the sake of clarity. We describe how to relax this assumption in Section 4, and work with datasets with missing preferences in Section 5.

**Collective Preferences as Partial Orders**   Standard approaches express collective preferences as a *ranking* – i.e., a total order over $n$ items where we can compare any pair of items. We consider an alternative approach in which we express collective preferences as a *tiered ranking*:

**Definition 2.1.** A *tiered ranking* $T$ is a partial ordering of $n$ items into $m$ disjoint *tiers* $T :=$ $(T_1, \ldots, T_m)$. Given a tiered ranking, we denote the collective preferences as:

$$
\pi_{i,j}(T) := \begin{cases}
1 & \text{if} \quad i \in T_l, j \in T_{l'} \ \text{for } l < l', \\
-1 & \text{if} \quad i \in T_l, j \in T_{l'} \ \text{for } l > l', \\
\bot & \text{if} \quad i, j \in T_l \ \text{for any } l
\end{cases}
$$

Tiers provide a way to *abstain from arbitration*. Given a pair of items where users disagree, we can place them in the same tier and "agree to disagree." Given a tiered ranking $T$, we can only make claims about collective preferences by comparing items in different tiers. In what follows, we say that a pairwise comparison between items $i, j$ is *valid* if $\pi_{i,j}(T) \neq \bot$. We refer to a valid pairwise comparison as a *selective comparison*.

**Selective Aggregation**   *Selective ranking* $S_\tau$ is a partial order that maximizes the number of comparisons that align with the preferences of at least $100 \cdot (1 - \tau)$ of users. Given a dataset of pairwise preferences over $n$ items from $p$ users, we can express $S_\tau$ as the optimal solution to an optimization problem over the space of all tiered rankings $\mathbb{T}$:

$$
\begin{aligned}
\max_{T \in \mathbb{T}} \quad & \text{Comparisons}(T) \\
\text{s.t.} \quad & \text{Disagreements}(T) \leq \tau p
\end{aligned}
\tag{$\text{SPA}_\tau$}
$$

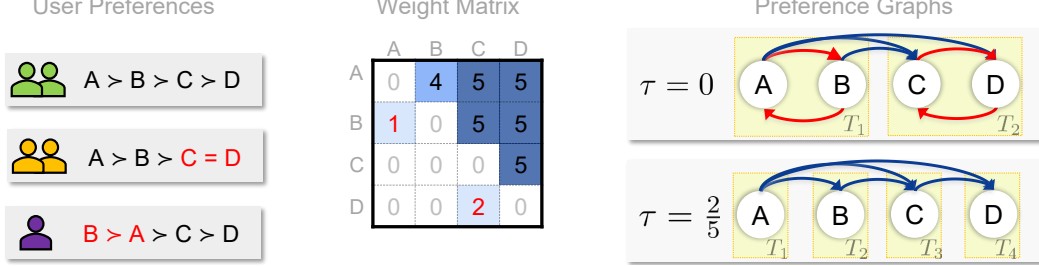

Figure 2: Graphical representations used to construct selective rankings for the preference aggregation task in Fig. 1. Here, the selective rankings for $\tau = 0$ and $\tau = \frac{2}{5}$ have 2 and 4 tiers, respectively.

Here, the objective maximizes the number of valid comparisons in a tiered ranking $T$:

$$\text{Comparisons}(T) := \sum_{i,j \in [n]} \mathbb{I}\left[\pi_{i,j}(T) \neq \bot\right]$$

The constraints restrict the fraction of individual preferences that can be contradicted by any valid comparison in $T$

$$\text{Disagreements}(T) := \max_{\substack{i,j \in [n] \\ k \in [p]}} \sum \mathbb{I}\left[\pi_{i,j}(T) = 1, \pi_{i,j}^k \neq 1\right]$$

The *dissent parameter* $\tau$ limits the fraction of individual preferences that can be violated by any selective comparison. Given a selective ranking $S_\tau$ that places item $i$ in a tier above item $j$, at most $100 \cdot \tau\%$ of users may have stated $i \not\succ j$.

We restrict $\tau \in [0, 0.5)$ to guarantee that the selective ranking $S_\tau$ aligns with a majority of users, and is unique (see Appendix B for a proof). In this regime, we can set $\tau$ to trade off coverage for alignment as shown in Fig. 4. Setting $\tau = 0$ returns a selective ranking that reflects unanimity by showing all comparisons on which all users agree. Setting $\tau$ just shy of 0.5 reflects a selective ranking that maximizes tiers without overruling a majority of users. The trade-off is analogous to the trade-off in selective classification [42–44]: we output a partial order (selective classifier) that sacrifices "comparisons" (coverage) to reduce "disagreement" (error).

## 3   Algorithms

We present an algorithm to construct selective rankings in Algorithm 1 and depict its behavior in Fig. 2.

---

**Algorithm 1** Selective Preference Aggregation

**Input:** $\{\pi_{i,j}^k\}_{i,j \in [n], k \in [p]}$                    *preference dataset*
**Input:** $\tau \in [0, 0.5)$                                        *dissent parameter*
1: $w_{i,j} \leftarrow \sum_{k \in [p]} \mathbb{I}\left[\pi_{i,j}^k \geq 0\right]$ for all $i, j \in [n]$
2: $V_I \leftarrow [n]$
3: $A_I \leftarrow \{(i \rightarrow j) \mid w_{i,j} > \tau p\}$
4: $V_T \leftarrow \mathsf{ConnectedComponents}(V_I, A_I)$
5: $A_T \leftarrow \{(T \rightarrow T') \mid \exists i \in T, j \in T' : (i \rightarrow j) \in A_I\}$
6: $l_1, \ldots, l_{|T|} \leftarrow \mathsf{TopologicalSort}(V_T, A_T)$
**Output:** $S_\tau \leftarrow (T_{l_1}, T_{l_2}, \ldots, T_{l_{|T|}})$          *$\tau$-selective ranking*

---

Algorithm 1 constructs a selective ranking from a dataset of pairwise preferences and a dissent parameter $\tau \in [0, 0.5)$. The procedure first builds a directed graph over items $(V_I, A_I)$. Here, each vertex corresponds to an item, and each arc corresponds to a collective preference that we must not contradict in a tiered ranking. Given $(V_I, A_I)$, the procedure then builds a directed graph over tiers $(V_T, A_T)$. In Line 4, it calls the $\mathsf{ConnectedComponents}$ routine to identify the strongly connected

components of $(V_I, A_I)$ which become the set of *supervertices* $V_T = \{T_1, \ldots, T_{|V_T|}\}$, where each supervertex contains items in the same tier. In Line 5, it defines arcs between tiers – drawing an arc from $T$ to $T'$ whose respective elements are connected by an arc in $A_I$. Given $(V_T, A_T)$, the procedure determines an ordering among tiers by calling the TopologicalSort routine in Line 6. In this case, the graph will admit a topological sort as it is a directed acyclic graph.

**Correctness** We show that Algorithm 1 recovers the unique optimal solution to $\mathsf{SPA}_\tau$ in Theorem B.2. The result follows from the fact that the directed graph $(V_T, A_T)$ defines a tiered ranking that is both feasible and optimal with respect to $\mathsf{SPA}_\tau$. Specifically, the tiered ranking must obey the disagreement constraint in $\mathsf{SPA}_\tau$ because we only draw arcs between items $i$ and $j$ that can violate the preferences of $\tau p$ users in Line 3. The tiered ranking maximizes the objective of $\mathsf{SPA}_\tau$ because the ConnectedComponents routine in Line 4 partitions vertices in a way that maximizes the number of tiers, which subsequently maximizes the selective comparisons under the disagreement constraint.

# 4 Theoretical Guarantees

In this section, we present formal guarantees on the stability and recovery of selective rankings.

**On the Recovery of Condorcet Winners** We often aggregate preferences to identify items that are collectively preferred to all others. Consider, for example, a task where we aggregate votes to select the most valuable player in a sports league or ratings to fund the most promising grant proposal [45]. In Theorem 4.1, we show that we can identify these "top" items from a solution path of selective rankings.

**Theorem 4.1.** *Consider a preference aggregation task where a majority of users prefer item $i_0$ to all other items. There exists a threshold value $\tau_0 \in [0, 0.5)$ such that, for every $\tau > \tau_0$, every selective ranking $S_\tau$ will place $i_0$ as the sole item in its top tier.*

Theorem 4.1 provides a formal recovery guarantee that ensures we recover a Condorcet winner or a Smith set [see e.g., 46] when they exist. In practice, the result implies that we can identify such "top items" by constructing and inspecting a solution path of selective rankings.

In tasks where a majority of users prefers an item to all others, the solution path will contain a selective ranking whose top tier consists of a single item. In this case, we can recover the "single winner" and report the threshold value $\tau_0$ as a measure of consensus.

In tasks where such a majority does not exist, every selective ranking $S_\tau$ for $\tau \in [0, 0.5)$ will include at least two items in the top tier. In settings where we aggregate preferences to identify a "single winner," we can point to the solution path as evidence that no such winner exists and use it as a signal that further deliberation is required [see e.g., 47].

**Stability with Respect to Missing Preferences** Standard methods can output rankings that change dramatically once we elicit missing preferences [48–50]. In Proposition 4.2, we show that we can build a selective ranking that abstains from unstable comparisons by setting missing preferences to $\pi_{i,j}^k = 0$.

**Proposition 4.2.** *Given a preference dataset with missing preferences $\mathcal{D}^{\mathrm{init}}$, let:*

- $\mathcal{D}^{\mathrm{true}} \supseteq \mathcal{D}^{\mathrm{init}}$ *be a complete dataset where we elicit missing preferences; and*

- $\mathcal{D}^{\mathrm{safe}} \supseteq \mathcal{D}^{\mathrm{init}}$ *be a complete dataset where we set missing preferences to $\pi_{i,j}^k = 0$.*

*For any dissent value $\tau \in [0, \frac{1}{2})$, let $S_\tau^{\mathrm{safe}}$ and $S_\tau^{\mathrm{true}}$ denote selective rankings for $\mathcal{D}^{\mathrm{safe}}$ and $\mathcal{D}^{\mathrm{true}}$, respectively. Then for any selective comparison $\pi_{i,j}(S_\tau^{\mathrm{safe}}) \in \{-1, 1\}$, we have:*

$$\pi_{i,j}(S_\tau^{\mathrm{safe}}) = \pi_{i,j}(S_\tau^{\mathrm{true}}).$$

Proposition 4.2 provides a simple way to ensure stability when working with datasets where we are missing preferences from certain users for certain items. In such cases, we can always build a $S$ that is "robust to missingness" in the sense that it will abstain from comparisons that may be invalidated once we elicit missing preferences.

**Stability with Respect to New Items**  In Proposition 4.3, we characterize the stability of selective aggregation as we add a new item to our dataset.

**Proposition 4.3.** *Consider a task where we start with a dataset of all pairwise preferences from $p$ users over $n$ items, which we then update to include all pairwise preferences for a new $n + 1^{th}$ item. For any $\tau \in [0, \frac{1}{2})$, let $S_\tau^n$ and $S_\tau^{n+1}$ denote selective rankings over $n$ items and $n + 1$ items, respectively. Then for any two items $i, j \in [n]$, we have:*

$$\pi_{i,j}(S_\tau^{n+1}) \in \{-1, 1\}, \pi_{i,j}(S_\tau^{n+1}) \neq -\pi_{i,j}(S_\tau^n)$$

The result shows that adding a new item to a selective ranking will either maintain each comparison or abstain. That is, adding a new item can only collapse items that were in different tiers into a single tier. However, it cannot lead items in the same tier to split. Nor can it lead items in different tiers to invert their ordering.

## 5   Experiments

In this section, we present an empirical study of selective aggregation on real-world datasets. Our goal is to benchmark the properties and behavior of selective rankings with respect to existing approaches in terms of transparency, robustness, and versatility. We include additional results in Appendix D, and code to reproduce our results on anonymized repository.

### 5.1   Setup

We work with 5 preference datasets from different domains listed in Table 1. Each dataset encodes user preferences over items as votes, ratings, or rankings. We convert preferences to pairwise comparisons with ties and build rankings using our approach and baselines. We construct solution paths using Algorithm 2 and report results for three dissent values:

- SPA$_0$: the selective ranking for $\tau = 0$. This solution reflects unanimous collective preferences.
- SPA$_{min}$: the selective ranking for the smallest positive dissent value $\tau > 0$ with 2+ tiers. This solution reflects the minimum disagreement we must incur to make any claim about collective preferences.
- SPA$_{maj}$: the selective ranking for the largest $\tau < 0.5$. This solution reflects the maximum number of claims we can make about collective preferences without overruling a majority of users.

We construct rankings using the following baseline methods:

- *Voting Rules*: We consider Borda [51] and Copeland [52], which are voting rules from social choice that rank items based on position or pairwise wins.
- *Sampling*: We use MC4 [2], which returns a ranking that orders items in terms of the stationary probabilities of a Markov chain where transitions are defined by random walks over user preferences.
- *Median Rankings*: We use Kemeny[53], which returns a ranking that minimizes collective disagreement. We report results for an exact approach that handles ties and returns a certifiably optimal ranking by solving an integer program using CPLEX v22 [54]. We report results using the BioConsert heuristic [55], which returns a ranking that minimizes collective disagreement through a local search heuristic.

### 5.2   Results

We summarize the specificity, disagreement, and robustness of rankings from all methods and all datasets in Table 1. In what follows, we discuss these results.

**On Transparency**  Some of the key issues with standard approaches stem from transparency. When we express collective preferences, we are forced to arbitrate disagreement yet unable to reveal information about arbitration. Given a ranking, we cannot tell how many users we had to overrule, which items were subject to conflicting preferences, or whether the collective preferences reflect genuine agreement or an artifact of forced ranking structure.

Our results highlight how selective rankings can address these issues on multiple fronts. As shown in Appendix D.1, a selective ranking can reveal the degree of disagreement through its dissent parameter,

| | | Selective | | | Standard | | | |
|---|---|---|---|---|---|---|---|---|
| **Dataset** | **Metrics** | SPA$_0$ | SPA$_{min}$ | SPA$_{maj}$ | Borda | Copeland | MC4 | Kemeny |
| nba
$n = 7$ items
$p = 100$ users
28.6% missing
NBA [56] | Disagreement Rate | 0.0% | 2.0% | 6.4% | 8.3% | 8.3% | 7.9% | 8.1% |
| | Abstention Rate | 100.0% | 42.9% | 28.6% | – | – | – | – |
| | # Tiers | 1 | 2 | 4 | 7 | 7 | 6 | 7 |
| | # Top Items | 7 | 3 | 1 | 1 | 1 | 1 | 1 |
| | $\Delta$ Sampling | 0.0% | 0.0% | 0.0% | 4.8% | 4.8% | 0.0% | 4.8% |
| | $\Delta$-Adversarial | 0.0% | 0.0% | 0.0% | 19.0% | 19.0% | 19.0% | 14.3% |
| survivor
$n = 39$ items
$p = 6$ users
0.0% missing
Purple Rock [57] | Disagreement Rate | 0.0% | 0.2% | 0.2% | 6.8% | 6.6% | 6.4% | 6.7% |
| | Abstention Rate | 94.9% | 42.5% | 42.5% | – | – | – | – |
| | # Tiers | 2 | 5 | 5 | 39 | 36 | 35 | 39 |
| | # Top Items | 1 | 1 | 1 | 1 | 1 | 1 | 1 |
| | $\Delta$ Sampling | 0.0% | 0.0% | 0.0% | 1.3% | 0.8% | 0.8% | 0.9% |
| | $\Delta$-Adversarial | 0.0% | 0.0% | 0.0% | 2.6% | 1.8% | 3.1% | 1.6% |
| lawschool
$n = 20$ items
$p = 5$ users
0% missing
LSData [58] | Disagreement Rate | 0.0% | 0.3% | 3.1% | 4.7% | 4.2% | 4.2% | 4.1% |
| | Abstention Rate | 40.5% | 36.8% | 4.2% | – | – | – | – |
| | # Tiers | 4 | 6 | 15 | 20 | 20 | 19 | 20 |
| | # Top Items | 12 | 12 | 2 | 1 | 1 | 1 | 1 |
| | $\Delta$ Sampling | 0.0% | 0.0% | 0.0% | 1.6% | 1.1% | 0.5% | 29.5% |
| | $\Delta$-Adversarial | 0.0% | 0.0% | 0.0% | 3.7% | 2.6% | 2.6% | 45.8% |
| csrankings
$n = 175$ items
$p = 5$ users
0.0% missing
Berger [59] | Disagreement Rate | 0.0% | 0.0% | 0.1% | 12.3% | 12.2% | 12.2% | 13.7%* |
| | Abstention Rate | 100.0% | 98.9% | 95.5% | – | – | – | – |
| | # Tiers | 1 | 2 | 3 | 175 | 168 | 170 | 175* |
| | # Top Items | 175 | 1 | 1 | 1 | 1 | 1 | 1* |
| | $\Delta$ Sampling | 0.0% | 0.0% | 0.0% | 0.8% | 0.8% | 0.1% | 9.0%* |
| | $\Delta$-Adversarial | 0.0% | 0.0% | 0.0% | 3.1% | 1.7% | 0.1% | 11.1%* |
| sushi
$n = 10$ items
$p = 5,000$ users
0.0% missing
Kamishima [60] | Disagreement Rate | 0.0% | 13.6% | 42.6% | 42.6% | 42.6% | 42.6% | 42.6% |
| | Abstention Rate | 100.0% | 64.4% | 0.0% | – | – | – | – |
| | # Tiers | 1 | 2 | 10 | 10 | 10 | 10 | 10 |
| | # Top Items | 10 | 8 | 1 | 1 | 1 | 1 | 1 |
| | $\Delta$ Sampling | 0.0% | 0.0% | 0.0% | 0.0% | 0.0% | 2.2% | 2.2% |
| | $\Delta$-Adversarial | 0.0% | 0.0% | 0.0% | 2.2% | 2.2% | 11.1% | 11.1% |

Table 1: Comparability, disagreement, and robustness of rankings for all methods on all datasets. We report the following metrics for each ranking: *Disagreement Rate*, i.e., the fraction of collective preferences that conflict with user preferences; *Abstention Rate*, i.e., the fraction of collective preferences that abstain from comparison; *# Tiers*, the number of tiers or ranks. *# Top Items*, i.e., the number of items in the top tier or rank. $\Delta$-*Sampling*, the average fraction of collective preferences that are inverted when we drop 10% of individual preferences; and $\Delta$-*Adversarial*, the maximum fraction of collective preferences that are inverted when we flip 10% of individual preferences, respectively.

and identify items where users disagree through its structure. Given a selective ranking, we are only allowed to compare items across tiers and are guaranteed that any comparison will overrule at most $\tau$ fraction of users. We can immediately tell that at least $\tau$ fraction of users express conflicting preferences over items within the same tier (e.g., Duke and Columbia).

In contrast to existing methods, selective aggregation only reveals a single winner or total order when collective preferences align with a majority of users. In Table 1, we see that preference aggregation tasks may not admit a single winner or a total order. In effect, we recover a selective ranking that identifies a single winner on 4 out of 5 datasets, and a total order on only 1 out of 5. In many cases, the inability to identify a winner or total order is meaningful. On the law dataset, for example, the most granular selective ranking SPA$_{maj}$ identifies two "top" schools (Stanford and Yale). On the sushi dataset, we recover selective rankings that identify both a single winner and a total order for $\tau = 0.48$. In practice, this means any ranking of population preferences over sushi is highly contentious.

**On Robustness** One of the main limitations of representing collective preferences as a ranking is that it may change dramatically as a result of changes to individual preferences [48–50]. This sensitivity is structural: given a ranking over $n$ items, changes to individual preferences can affect any of the $\binom{n}{2}$ pairwise preferences. In contrast, selective rankings limit sensitivity by grouping items into $m \leq n$ tiers, which can induce robustness by restricting the number of comparisons that are subject to change.

In Table 1, we highlight this behavior by reporting the expected number of collective preferences that are inverted when we build a ranking from a dataset with a small number of missing or noisy preferences. Specifically, we report $\Delta$-Sampling and $\Delta$-Adversarial which measure the expected rate of inversions from sampling or gaming. Given each dataset and each method, we construct these estimates by applying each method to build a modified ranking from a dataset where we drop or flip 10% of individual preferences. We repeat this process 100 times and measure the number of collective preferences that change between the original ranking and the ranking we obtain for the modified dataset.

**On the Arbitrariness of Arbitration** Although methods for preference aggregation must arbitrate conflicting preferences, many are not built to arbitrate these differences with explicit guarantees. In Table 1, only SPA and Kemeny can pair rankings with formal guarantees on the arbitration process.

Kemeny can return a ranking that is guaranteed to minimize collective disagreement by solving a combinatorial optimization problem. In our experiments, we are able to recover a certifiably optimal ranking quickly for 4/5 datasets using a commercial solver on a single-core CPU with 128GB RAM. On the `csrankings` dataset, however, we are forced to use a heuristic approach [55] because the optimization problem contains a prohibitively large number of constraints.

## 6 Learning by Agreeing to Disagree

One of the most common use cases for preference aggregation in machine learning arises when we align models with the collective preferences of their users. – e.g., to predict the toxicity of an online comment or the helpfulness of a chatbot response. In the simplest case, we would recruit users to annotate a set of training examples. We would then aggregate their annotations to obtain aggregate training labels we could use to fit or fine-tune a model [61]. We often apply this pattern in tasks where we wish to predict outcomes where individuals express conflicting preferences due to ambiguity [26] or subjectivity [20, 30]. In such cases, standard aggregation methods such as majority vote can lead to models whose predictions reflect the collective preferences of the majority [3, 12]. In what follows, we explore how selective aggregation can mitigate these effects by returning training labels that better account for all annotators' views.

**Setup** We consider a task to build a classifier to detect toxic conversations with a language model. We work with the DICES dataset [62], which contains individual toxicity labels for $n = 350$ chatbot conversations from $p = 123$ users. Here, each label is defined as $y_i^k \in \{1, -1, 0\}$ if user $k$ labels conversation $i$ as {toxic, benign, unsure} respectively. We randomly split users into two groups: a group of $p^{\text{train}} = 5$ users whose labels we use to train our model; and a group of $p^{\text{test}} = 118$ users whose labels we use to evaluate the predictions of the model at an individual level once it is deployed. We set the relative size of each group to reflect a practical setting where a company would collect labels from a small subset of users to train a model that assigns predictions to a large population.

We aggregate the toxicity labels from each user in the training set to create three sets of aggregate training labels to train our model. In this case, we drop all annotations where a user rates a conversation as "unsure" – i.e., where $y_{i,k} = 0$ – and only aggregate annotations for conversations that are labeled as toxic or non-toxic – i.e., $y_{i,k} \in \{-1, 1\}$.

- $y_i^{\text{Maj}} := \mathbb{I}\left[\sum_{k \in [p]} \mathbb{I}\left[y_i^k = 1\right] \geq \sum_{k \in [p]} \mathbb{I}\left[y_i^k = -1\right]\right]$, which denote aggregate labels from majority vote [63].
- $y_i^{\text{Borda}} \in [280]$, which denote aggregate labels from a pairwise variant of Borda [64]. As rankings are not provided, an item's score is calculated as its total number of pairwise wins, summed across all users.
- $y_i^{\text{SPA}} \in [15]$, which denotes aggregate labels from SPA for the large dissent parameter $\tau < 0.5$.
- $y_i^{\text{Exp}} \in [4]$, which denote ratings elicited from an in-house expert. This reflects a baseline where we train a model using annotations from a single human expert.

We process the training labels from each method to ensure that we can use a standard training procedure across similar methods. We use the training labels from each method to fine-tune a BERT-Mini model [65] and denote these models as $f^{\text{SPA}}, f^{\text{Maj}}, f^{\text{Borda}}, f^{\text{Expert}}$. We evaluate how each method performs with respect to individuals and users in a specific group in terms of the following measures:

$$\text{BER}_k(f^{\text{all}}) := \tfrac{1}{2}\text{FPR}_k(f^{\text{all}}) + \tfrac{1}{2}\text{FNR}_k(f^{\text{all}})$$

$$\text{LabelError}(y^{\text{all}}) := \tfrac{1}{p}\sum_{k=1}^{p}\text{BER}_k(y^{\text{all}})$$

$$\text{PredictError}(f^{\text{all}}) := \tfrac{1}{p}\sum_{k=1}^{p}\text{BER}_k(f^{\text{all}})$$

We evaluate the performance of each in terms of the balanced error rate for clarity as the data for each user exhibits class imbalance that changes across users. We include additional details on our setup in Appendix D.5.

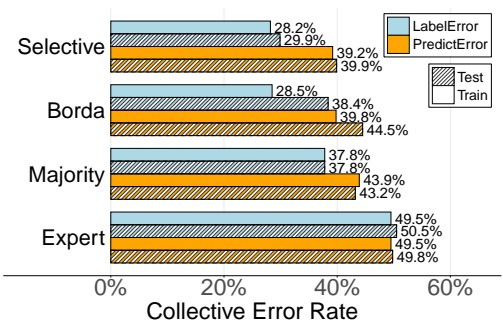
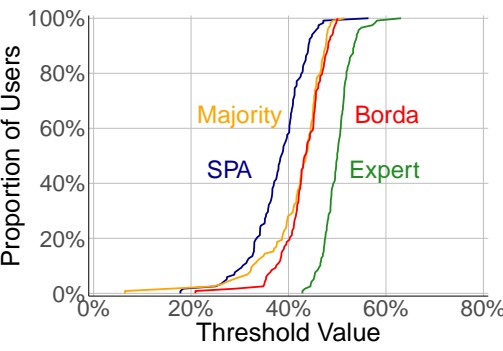

(a) Group-level errors on DICES. LabelError = avg. disagreement with annotators; PredictError = avg. disagreement of model predictions. Train: $p^{\text{train}}{=}5$; Test: $p^{\text{test}}{=}118$. SPA is lowest on both.

(b) CDF of user-level BER on held-out users ($p^{\text{test}}{=}118$). Higher is better.

**Results**     We summarize our results at a group level and individual level in Section 6.

Our results highlight how SPA aggregates labels in a way that minimizes collective disagreement – achieving a label error of 28.2% (c.f. 37.8% with $y^{\text{Maj}}$). Moreover, the improved alignment in training labels can propagate into an improved alignment in the predictions of the model. In this case, $f^{\text{SPA}}$ has a prediction error of 29.9% on training users and 39.9% on test users (c.f. 38.4 % and 44.5 % for $f^{\text{Borda}}$).

We also show how the prediction error is distributed across the $p^{\text{train}} = 5$ annotators in the train set – i.e., users whose preferences we would typically observe, as well as the $p^{\text{test}} = 118$ held-out annotators, whose preferences we would not typically be able to observe. In this case, roughly 60% of users achieve an individual BER of 40% or less under $y^{\text{SPA}}$, compared to roughly 20% of users for $y^{\text{Borda}}$ and $y^{\text{Maj}}$.

Our results highlight a benefit from building models using labels that encode collective preferences. In this case, the large values of label error for $y^{\text{Exp}}$ imply that many users disagree with the expert. These findings capture the performance of each approach in a task where we threshold the predictions of each method to optimize the BER. In practice, we observe similar findings at other salient operating points – e.g., requiring a collective TPR of $\geq 90\%$. In such cases, baselines such as majority vote may underperform as their labels can only capture binary information.

# 7   Concluding Remarks

In many applications where we aggregate human preferences, disagreement is "signal, not noise" [18]. In this work, we developed foundations to aggregate preferences in a way that can reveal disagreement and allow us to control it. Selective aggregation compares only on consensus, resisting adversarial flips and missing data by abstaining on contested pairs. The main limitation of our work stems from algorithm design: the algorithms we have developed in this work are designed to be simple, versatile, and safe. To this end, they behave conservatively in tasks where datasets contain a large number of missing preferences. Such datasets are common in tasks where it is costly to elicit preferences or where we must elicit preferences over a large collection of items. In these cases, we can still represent collective preferences as a selective ranking, but the output may collapse into a single tier. This behavior is intentional: it signals that any claim about the collective preferences could be invalidated once the missing preferences are elicited. At the same time, it is impractical in large-scale applications that rely on sparse data and elicit only a few preferences from each user. Looking forward, we can extend our paradigm to such settings by adopting probabilistic assumptions [see e.g., 39] and by developing procedures to streamline preference elicitation.

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

# Supplementary Materials
# Selective Preference Aggregation

# A  Supplementary Material for Section 2

## A.1  Notation

We provide a list of the notation used throughout the paper in Table 2.

| Object | Symbol | Description |
|---|---|---|
| Items | $i \in [n] := \{1, \ldots, n\}$ | The objects being ordered, for which users have expressed preferences. |
| Users | $k \in [p] := \{1, \ldots, p\}$ | Individuals expressing preferences for given items. |
| Individual preferences | $\pi_{i,j}^{k} \in \{-1, 0, 1\}$ | Pairwise preference between items $i$ and $j$ for user $k$. |
| Tiered ranking | $T$ | A partial ordering of $n$ items into $m$ *tiers* |
| Collective preference | $\pi_{i,j}(T) \in \{-1, 0, 1\}$ | The preference between items $i$ and $j$ in a given ranking. |
| Selective ranking | $S_\tau$ | The partial order returned by solving $\mathsf{SPA}_\tau(\mathcal{D})$. |
| Dissent parameter | $\tau \in [0, \frac{1}{2})$ | The admitted dissent between two items $i$ and $j$. |

Table 2: Notation

## A.2  Encoding Individual Preferences as Pairwise Comparisons

| Representation | Notation | Conversion |
|---|---|---|
| Labels | $y_i^k \in \{0, 1\}$ | $\pi_{i,j}^k = \mathbb{I}\left[y_i^k > y_j^k\right] - \mathbb{I}\left[y_i^k < y_j^k\right]$ |
| Ratings | $y_i^k \in [m]$ | $\pi_{i,j}^k = \mathbb{I}\left[y_i^k > y_j^k\right] - \mathbb{I}\left[y_j^k > y_i^k\right]$ |
| Rankings | $r^k : [n] \to [n]$ | $\pi_{i,j}^k = \mathbb{I}\left[r^k(i) > r^k(j)\right] - \mathbb{I}\left[r^k(i) < r^k(j)\right]$ |

Table 3: Data structures that capture ordinal preferences over $n$ items. Each representation can be converted into a set of $\binom{n}{2}$ pairwise preferences in a way that ensures (and assumes) transitivity. Item-level representations require fewer queries but may be subject to calibration issues between annotators.

One of the benefits in developing machinery to aggregate preferences is that it can provide practitioners with flexibility in deciding how to elicit and aggregate the preferences. In practice, such choices involve trade-offs that we discuss briefly below. Specifically, eliciting pairwise preferences from users requires more queries than other approaches [66]. However, it may recover a more reliable representation of ordinal preferences than ratings or rankings [i.e., 67]. In tasks where we work with a few items, we can elicit preferences as ratings, rankings, or pairwise comparisons. In tasks where we elicit rankings, we can convert them into pairwise comparisons without a loss of information. In this case, eliciting pairwise comparisons can test implicit assumptions such as transitivity. In tasks where we elicit labels and ratings, the conversion is lossy – since we are converting cardinal preferences to ordinal preferences. In practice, this conversion can resolve issues related to calibration across users [see e.g, 68, 69]. In theory, it may also resolve disagreement [34].

# B Supplementary Material for Section 3

This appendix provides supplementary material for Section 3, including proofs of the claims in this section and a description of the solution path algorithm.

## B.1 Proof of Correctness

**Lemma B.1.** Consider the graph before running condensation or topological sort, but after pruning edges with weights below $\tau p$. Items can be placed in separate tiers without violating Disagreements$(T) \leq \tau p$ if and only if there is no cycle in the graph involving those items.

*Proof.* We start by connecting the edges in a graph to conditions on the items in a tiered ranking and eventually expand that connection to show the one-to-one correspondence between cycles and tiers.

First note that for any items $i, j$: $w_{i,j} > \tau \iff \sum_{k=1}^{p} 1\left[\pi_{i,j}^{k} \neq 1\right] > \tau p$. This follows trivially from the definition of $w_{i,j} := \sum_{k=1}^{p} 1\left[\pi_{i,j}^{k} \neq 1\right]$. From this, we know that if and only if there exists an arc $(i, j)$ that is not pruned before condensation, we cannot have a tiered ranking with $\pi_{i,j}^{T} = -1$ without violating Disagreements$(T) \geq \tau p$.

If there exists a cycle in this graph, then we know the items in that cycle must be placed in the same tier. To show this, consider some edge $i, j$ in the cycle. We know item $j$ cannot be in a lower tier than $i$ without violating the disagreements property, from the above. So item $j$ must be in the same or a higher tier. But item $j$ has an arrow to another item, $k$, which must be in the same or a higher tier than both $j$ and $i$, and so on, until the cycle comes back to item $i$. This corresponds to the constraint that all items must be in the same tier.

If a set of items is not in a cycle, then these items do not need to be placed in the same tier. If the items are not in a cycle, then there exists a pair of items $(i, j)$ such that there is no path from $j$ to $i$. Thus $i$ can be placed in a higher tier than $j$ without violating any disagreement constraints. Thus not all items in this set need to be placed in the same tier.

Thus we have shown that for a graph pruned with a given value of $\tau$, items can be placed in separate tiers for a tiered ranking based on that same parameter $\tau$, if and only if there is no cycle in the graph involving all of these items.

$\square$

We draw on this Lemma to prove the main result:

**Theorem B.2.** Given a preference aggregation task with $n$ items and $p$ users, Algorithm 1 returns the optimal solution to $\mathsf{SPA}_\tau$ for any dissent parameter $\tau \in [0, \frac{1}{2})$.

*Proof of Theorem B.2.* Consider that items in our solution are in the same tier if and only if they are part of a cycle in the pruned graph (i.e., if and only if they are in the same strongly connected component). So items are in the same tier if and only if they must be in the same tier for the solution to be feasible. No other feasible tiered ranking could have any of these items in separate tiers. So no other tiered ranking could have any more tiers, or any more comparisons. To do so would require placing some same-tier items in different tiers. Thus, our solution is maximal with respect to the number of tiers, and with respect to the number of comparisons. $\square$

## B.2 Proof of Uniqueness

**Theorem B.3.** The optimal solution to $\mathsf{SPA}_\tau$ is unique for $\tau \in [0, 0.5)$.

*Proof of Theorem B.3.* Let $T$ denote an optimal solution to $\mathsf{SPA}_\tau$. We will show that the optimality $T$ is fully specified by: (1) the items in each tier and (2) the ordering between tiers. That is, if we were to produce a tiered ranking $T'$ that assigns different items to each tier, or that orders tiers in a different way would be suboptimal or infeasible.

Consider a tiered ranking $T$ that is feasible with respect to $\mathsf{SPA}_\tau$ for some $\tau \in [0, 0.5)$. Let $T'$ denote a tiered ranking where we swap the order of two tiers in $T$. We observe that the $T'$ must violate a

constraint. To see this, consider any pair of items $i, j$ such that $\pi_{i,j}(T) = 1$ before the swap, but $\pi_{j,i}(T') = 1$ after the swap. One such pair must exist for any swapping of tier orders, because all tiers are non-empty. Because we elicited complete preferences, one of the following conditions must hold:

$$\sum_{k \in [p]} \mathbb{I}\left[\pi_{i,j}^k \neq 1\right] > \tau p \tag{1}$$

$$\sum_{k \in [p]} \mathbb{I}\left[\pi_{j,i}^k \neq 1\right] > \tau p \tag{2}$$

Assuming that $T$ was an optimal solution to $\mathsf{SPA}_\tau$, we observe that the condition in Eq. (1) must be violated because the original optimal solution was valid. Thus, we must have that $\sum_{k \in [p]} \mathbb{I}\left[\pi_{j,i}^k \neq 1\right] > \tau p$. This implies that $\text{Disagreements}(T') > \tau p$ for this tiered ranking. Thus, swapping the order of tiers violates constraints because $\tau < 0.5$.

Now note that any separation of items from within the same tier is not possible without violating a constraint. This follows from Lemma B.1, which states that items that are part of a cycle in our graph representation of the problem[1], must be in the same tier for a solution to be valid. And, as specified in our algorithm, we know our optimal solution has tiers only where there are cycles in the graph representation of the problem. So any tiers in the optimal solution cannot be separated.

We can still merge two tiers together without violating constraints, but such an operation reduces the number of comparisons and would no longer be optimal. And after merging two tiers, the only valid separation operation would be simply to undo that merge (since any other partition of the items in that merged tier, would correspond to separating items that were within the same tier in the optimal solution). So we cannot use merges as part of an operation to reach a valid alternative optimal solution.

So we know that for the optimal solution, we cannot separate out any items within the same tier, and we cannot reorder any of the tiers. Merging, meanwhile, sacrifices optimality. Thus, the original optimal solution is unique. $\qquad\square$

## B.3 Constructing All Possible Selective Rankings

We start with a proof for Proposition B.5.

*Proof of Proposition B.5.* Recall that in Algorithm 1, an edge $(i, j)$ with weight $w_{i,j}$ is excluded if at least $\tau p$ users disagree with the preference $j \succ i$. We observe that $w_{i,j} = \sum_{k \in [p]} \mathbb{I}\left[\pi_{i,j}^k \geq 0\right]$ corresponds the number of users who disagree with the preference $j \succ i$. Given a dataset, denote the set of dissent values that could lead to different outputs as:

$$\mathcal{W} = \{0\} \cup \left\{ \tau' \mid \exists i, j : \tau' = \left( \frac{1}{p} \sum_{k \in [p]} \mathbb{I}\left[\pi_{i,j}^k \geq 0\right] \right) < \tfrac{1}{2} \right\}$$

This corresponds to the set of unique $w_{i,j}/p$ for all $i, j$, with the value 0 included as well. To see this, note $w_{i,j} = \sum_{k \in [p]} \mathbb{I}\left[\pi_{i,j}^k \geq 0\right]$. We will now show the following Lemma, which will resolve the original claim.

**Lemma B.4.** *Given any two adjacent elements $a, b \in \mathcal{W} \cup \{\tfrac{1}{2}\}$. All dissent values in $\tau \in [a, b)$ lead to the same selective ranking as the selective ranking for $\tau = a$.*

*Proof.* To show this, note that there exists no edge $i \to j$ such that $ap < w_{i,j} < bp$. If there did exist, then we would have

$$a < \frac{w_{i,j}}{p} < b.$$

This would imply that $\mathcal{W}$ would have to include an additional between $a$ and $b$. But $a$ and $b$ are adjacent in $\mathcal{W}$. This is a contradiction.

---

[1]after pruning edges of weight below $\tau$

657 Since there exists no edge $i \to j$ such that $ap < w_{i,j} < bp$, there exists no edge such that the decision
658 to include its arc in the graph changes based on what value of dissent we select in $[a, b)$. Recall that
659 we exclude $i \to j$ iff $w_{ij} \geq \tau p$ $\qquad\square$

660 Now that we know that for any two adjacent values a, b in $\mathcal{W} \cup \{\frac{1}{2}\}$, all dissent values in $[a, b)$ lead
661 to the same tiered ranking as with dissent value $a$, we know that for any dissent value $\tau \in [0, \frac{1}{2})$, the
662 largest value of $\tau' \in \mathcal{W}$ that is $\leq \tau$ will lead to the same tiered ranking. Simply substitute $\tau$ in for $a$,
663 and the smallest value above $\tau$ in $\mathcal{W} \cup \{\frac{1}{2}\}$ for $b$ (such a value exists, on both sides, because 0 and $\frac{1}{2}$
664 are both $\in \mathcal{W} \cup \{\frac{1}{2}\}$, and $\tau \in [0. \frac{1}{2}))$.

665 Thus we have shown the required claim.

666 $\qquad\square$

667 **Recovering All Selective Rankings** Algorithm 1 is meant to recover a selective ranking in settings
668 where we can set the value of $\tau$ a priori (e.g., $\tau = 0\%$ to enforce unanimity). In many applications,
669 we may wish to set $\tau$ after seeing the entire path of selective rankings. In a funding task where we
670 only have the resources to fund 3 proposals, for example, we can choose the smallest value of $\tau$ from
671 the solution path such that the top tier contains $\leq 3$ proposals. In cases where a top three does not
672 exist, this can lead us to save resources or increase our budget. In a prediction task where labels
673 encode collective preferences, we could aggregate annotations with a selective ranking and treat $\tau$ as
674 a hyperparameter to control overfitting.

675 In these situations, we can produce a *solution path* of selective rankings– i.e., a finite set of selective
676 rankings that covers all possible solutions to $\mathsf{SPA}_\tau$ for $\tau \in [0, \frac{1}{2})$ [c.f. 70]. We observe that a finite
677 solution path must exist as each selective ranking is specified by the arcs in Line 3. In practice, we can
678 compute all selective rankings efficiently by: (1) identifying a smaller subset of dissent parameters to
679 consider as per Proposition B.5; and (2) re-using the graph of strongly connected components across
680 iterations.

681 **Proposition B.5.** *Given a dataset of pairwise preferences $\mathcal{D}$, let $\mathcal{S}_\mathcal{W}$ denote a finite set of selective*
682 *rankings for dissent parameters in the set:*

$$\mathcal{W} = \left\{ \frac{w}{p} < \frac{1}{2} \mid w = \sum_{k \in [p]} \mathbb{I}\left[\pi_{i,j}^k \geq 0\right] \text{ for } i, j \in [n] \right\} \cup \{0\}$$

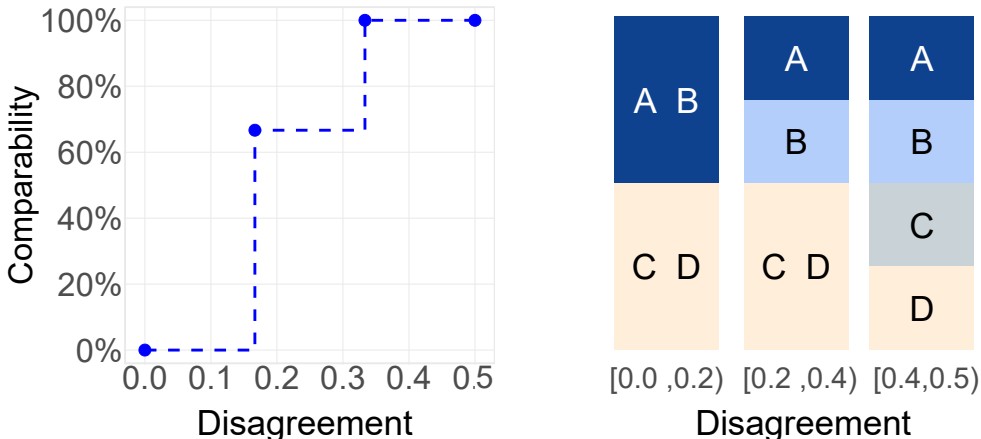

Figure 4: All possible selective rankings for the task in Fig. 1 where we aggregate the preferences
of $p = 5$ users over $n = 4$ items $\{A, B, C, D\}$. We show the comparability and disagreement of
each solution to $\mathsf{SPA}_\tau$ on the left, and their selective rankings on the right. Here, the solution for
$\tau \in [0, \frac{1}{5}]$ reveals that all users unanimously prefer $\{A, B\}$ to $\{C, D\}$. The solution for $\tau \in (\frac{1}{5}, \frac{2}{5}]$,
reveals that we can recover a single winner if we are willing to make claims that overrule at most 1
user, while the solution for $\tau \in (\frac{2}{5}, \frac{1}{2}]$ reveals we can only recover a total order if we are willing to
overrule at most 2 users.

*Let $S_\tau$ be a selective ranking for an arbitrary dissent value $\tau \in [0, \frac{1}{2})$. Then, $\mathcal{S}_\mathcal{W}$ contains a selective ranking $S_{\tau'}$ such that $S_{\tau'} = S_\tau$ for some dissent value $\tau' \leq \tau$.*

We describe this procedure in Algorithm 2. Both Algorithms 1 and 2 run in time $\mathcal{O}(n^2 p)$ – i.e., they are linear in the number of individual pairwise preferences elicited (see Appendix B.4). As we show in Fig. 5, the resulting approach can lead to an improvement in runtime in practice.

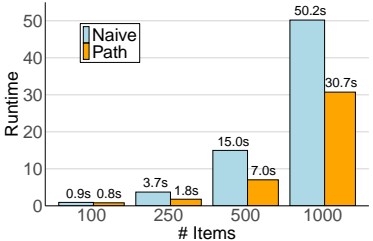

Figure 5: Runtimes to produce all selective rankings for a synthetic task with $p = 10$ users and $n$ items (see Appendix B for details). We show results for a naïve approach where we call Algorithm 1 for all possible dissent values, and the solution path algorithm in Appendix B. All results reflect timings on a consumer-grade CPU with 2.3 GHz and 16 GB RAM.

**Algorithm** We present an algorithm to construct a solution path of selective rankings in Algorithm 2.

---

**Algorithm 2** Solution Path Algorithm

---

**Input:** $\mathcal{D} = \{\pi^k_{i,j}\}_{i,j \in [n], k \in [p]}$ *preference dataset*
1: $\mathcal{S} = \{\}$ *initialize solution path*

  *Construct Initial Preference Graph for $\tau = 0$*

2: $w_{i,j} \leftarrow \sum_{k \in [p]} \mathbb{I}\left[\pi^k_{i,j} \geq 0\right]$ for all $i, j \in [n]$   $w_{i,j} = \#$ *preferences claiming $i \succeq j$*
3: $V_I \leftarrow [n]$ *Vertices represent items*
4: $A_I \leftarrow \{(i \to j) \mid w_{i,j} \geq 0\}$ *Arcs for observed preferences*

  *Construct Selective Rankings for All Possible Dissent Values*

5: $\mathcal{W} \leftarrow \{w_{i,j} \text{ for all } i, j \in [n] \mid w_{i,j} < \lceil \frac{p}{2} \rceil\} \cup \{0\}$ *Set of dissent parameters (see Proposition B.5)*
6: **for** $\tau \in \mathcal{W}$ **do**
7: $\quad A_I \leftarrow A_I / \{(i \to j) \in \mid w_{i,j} \geq \tau p\}$ *Add arcs with support $\geq \tau p$*
8: $\quad V_T \leftarrow \mathsf{ConnectedComponents}((T, A_T))$ *Group items into tiers*
9: $\quad A_T \leftarrow \{(T \to T') \mid \exists i \in T, j \in T' : (i \to j) \in A_I\}$ *Add edges between items to supervertex*
10: $\quad (l_1, \ldots, l_{|V_T|}) \leftarrow \mathsf{TopologicalSort}((V_T, A_T))$ *Sort components based on directed edges*
11: $\quad S_\tau \leftarrow (T_{l_1}, \ldots, T_{l_{|V_T|}})$
12: $\quad \mathcal{S} \leftarrow \mathcal{S} \cup \{S_\tau\}$
13: **end for**
**Output:** $\mathcal{S}$ *Selective rankings that cover the comparison-disagreement frontier*

---

Given a preference dataset Algorithm 2 returns a finite collection of selective rankings $\mathcal{S}$ that achieve all possible trade-offs of comparability and dissent. The procedure improves the scalability by restricting the values of the dissent parameter $\tau$ as per Proposition B.5 in Line 2, and by reducing the overhead of computing graph structures. In this case, we construct the preference graph once in Line 4, and progressively add arcs with sufficient support in Line 7.

Algorithm 2 assumes a complete preference dataset – i.e., where we have all pairwise preferences from all users. In practice, we can satisfy this assumption by imputing missing preferences to 0 as described in Proposition 4.2. Alternatively, we can also add an additional step after Line 7 to check that the item graph $(V_I, A_I)$ remains connected.

**Details on Synthetic Dataset in Fig. 5** We benchmarked Algorithm 2 against Algorithm 1 in Fig. 5 on synthetic preference aggregation tasks where we could vary the number of users and items.

We fixed the number of users to $p = 10$ users. For each user $k \in [p]$, we sampled their pairwise preferences as $\pi_{i,j}^k \sim \mathsf{Uniform}(1, 0, -1)$.

## B.4 Proofs of Algorithm Runtime

**Algorithm 1**   Line 1 computes a sum while visiting each pairwise preference for each judge, taking $\mathcal{O}(n^2 p)$ time. All subsequent steps are linear in the graph size: both ConnectedComponents and TopologicalSort are linear in input size, and the other steps are just operations on each edge. So the total runtime is $\mathcal{O}(n^2 p)$.

**Algorithm 2**   Note that $|\mathcal{W}| = \lceil \frac{p}{2} \rceil$, because $w_{ij}$ only takes integer values and there are $\lceil \frac{p}{2} \rceil$ integers between 0 and $\lceil \frac{p}{2} \rceil$ inclusive of 0 and exclusive of $\lceil \frac{p}{2} \rceil$. so the for loop runs $\lceil \frac{p}{2} \rceil$ times, and everything in the loop runs in time linear in the graph size, so $\mathcal{O}(n^2)$. Thus the whole runtime of the loop is $\mathcal{O}(n^2 p)$. The preprocessing, as before, is $\mathcal{O}(n^2 p)$ time. Note that computing $\mathcal{W}$ can be done in $\mathcal{O}(n^2 p)$ time: just iterate through all $w_{ij}$ for each of the $\lceil \frac{p}{2} \rceil$ possible distinct values, and add the value to $\mathcal{W}$ if it occurs at least once. Thus the total runtime is the sum of a constant number of $\mathcal{O}(n^2 p)$ steps, meaning the total runtime is $\mathcal{O}(n^2 p)$.

 # C    Supplementary Material for Section 4

 This appendix provides proofs and additional results to support the claims in Section 4.

 ## C.1    On the Top Tier

**Theorem C.1.** *Consider a preference aggregation task where at most $\alpha < \frac{1}{2}$ of users strictly prefer one item over all other items. Given any $\tau \in [0, \frac{1}{2})$, the tiered ranking from $\mathsf{SPA}_\tau$ will include at least two items in its top tier.*

*Proof.* We show the contrapositive: having $> (1 - \tau)$ users rank an item first guarantees having only one item in the top tier. Without loss of generality, call an item with $> (1 - \tau)$ users rating a specific item first $A$. Consider WLOG any other item $B$. No more than $\tau$ users claim either of $B \succ A$ or $B \sim A$, because we know $> (1 - \tau)$ users claim $A \succ B$. So for any tiered ranking that places some other item $B$ in the same tier as $A$, we could instead place $A$ above all other items in that tier, and have one more item. Since the result of our algorithm must have the maximal number of tiers, we cannot have a case where $A$ is in the same tier as any other item.     $\square$

**Lemma C.2.** *Consider a preference aggregation task where a majority of users strictly prefer an item $i_0$ over all items $i \neq i_0$. There exists some threshold dissent $\tau_0 \in [0, \frac{1}{2})$ such that for all $\tau > \tau_0$, every selective ranking we obtain by solving $\mathsf{SPA}_\tau$ will place $i_0$ as the sole item in its top tier.*

*Proof.* Let $\alpha$ denote the fraction of users who strictly prefer $i_0$ over all items. Since $\alpha > \frac{1}{2}$, we observe that at most $1 - \alpha < 1 - \frac{1}{2}$ users can express a conflicting preference. Given any item $i \neq i_0$, let $\tau_0 = 1 - \alpha$ denote the fraction who users who believe either of $i \succ i_0$ or $i \sim i_0$. For any tiered ranking that places $i_0$ and $i$ in the same tier, we could instead place $i$ above all other items in that tier, and have one more tier. Since our algorithm returns a tiered ranking with the maximal number of tiers, we cannot have a case where $i$ is in the same tier as any other item.     $\square$

 ## C.2    On Missing Preferences

*Proof of Proposition 4.2.* If we are missing preferences, our algorithm's behavior is to assume all missing preferences would be in disagreement with any asserted ordering. This exactly corresponds to the actual disagreement if the true values are all asserted equivalence/indifference, and an upper bound on dissent if the preferences are directional. By doing this, we guarantee that the disagreement property will be satisfied under any possible missingness mechanism, even a worst-case adversarial mechanism. We denote missingness as $\pi_k(i, j) = ?$ if the preference is missing. This property is trivial to show. Consider that

$$
\begin{aligned}
\text{Disagreements}(T) := \max_{\substack{i,j \in T, T' \\ T \succ T'}} \sum_{k \in [p]} \mathbb{I}\left[\pi_{i,j}^k \neq 1\right] \\
\leq \max_{\substack{i,j \in T, T' \\ T \succ T'}} \sum_{k \in [p]} 1\left[\pi_{i,j}^k \in \{0, -1, ?\}\right] \\
= \max_{\substack{i,j \in T, T' \\ T \succ T'}} \sum_{k \in [p]} \mathbb{I}\left[\pi_{i,j}^k \in \{0, -1\}\right] \text{ if we we set all missing values } \pi_{i,j}^k = ? \text{ to } \pi_{i,j}^k = 0
\end{aligned}
$$

Given that overall disagreement when preferences are imputed cannot increase, we have that $\pi_{i,j}(S_\tau^{\text{true}}) = \pi_{i,j}(S_\tau^{\text{safe}})$.

More formally: from the disagreements argument above, we know that $\mathcal{D}^{\text{safe}}$ has the same or more disagreements for any preference than does $\mathcal{D}^{\text{true}}$. Every selective comparison in $S_\tau^{\text{safe}}$ corresponds to a pair of items in distinct strongly connected components under the constraints from $\mathcal{D}^{\text{safe}}$ (see Lemma B.1). When we relax to only the constraints from $\mathcal{D}^{\text{true}}$, we cannot have more disagreement for any preferences, so those items will remain in distinct strongly connected components. Since they remain in distinct strongly connected components, Lemma B.1 tells us the two items will not be in the same tier.

To show that the two items will have the same ordering in both tiered rankings, note that even under $\mathcal{D}^{\text{true}}$ there must be a constraint on one of the two directions of the preference[2]. And that constraint will still hold under $\mathcal{D}^{\text{safe}}$, which is no less constrained than $\mathcal{D}^{\text{true}}$. Thus, $S_\tau^{\text{true}}$ cannot have a preference in the opposite direction from $S_\tau^{\text{safe}}$

$\square$

## C.3 On the Distribution of Dissent

A selective ranking only allows comparisons that violate at most $\tau p$ of preferences in a dataset. In practice, these violations may be disproportionately distributed across users or items. For example, we may have a task with $\tau = \frac{1}{p}$ where the same user disagrees with all comparisons in a dataset. Alternatively, the violations may be equally distributed across users – so that there is no coalition of users who agrees with all preferences. In Remark C.3, we bound the number of users who can disagree with a selective ranking.

*Remark* C.3. A $\tau$-selective ranking contradicts the preferences of at most $\frac{p^2}{4} \cdot \tau p$ users.

The result in Remark C.3 only applies in tasks where the number of users exceeds the number of selective comparisons. In other tasks – where the number of selective comparisons exceeds the number of users – the statement is vacuous as we cannot rule out a worst-case where every user disagrees with at least one comparison.

*Proof.* We observe that a selective ranking with a single tier makes no claims. Thus we can restrict our attention to cases where the $\tau$-selective ranking contains at least two tiers. Given a selective ranking with more than 2 tiers, then any user who disagrees with the ranking of items from non-adjacent tiers, also disagrees with the ranking of two items in adjacent tiers. So every user with a conflict must disagree about the ordering of at least one pair of items in adjacent tiers. This bounds the number of users who disagree as $\tau$ times the number of distinct pairs of items in adjacent tiers. This is because no more than $\tau$ proportion of users can disagree with any one pairing.

The number of distinct, adjacent-tier pairs is of the form $\sum_{l=1}^{|T|-1} n_l n_{l+1}$ where tier ; contains $n_l$ items, and all the tiers together contain all $n$ items ($\sum_{i=l} |T| n_l = n$). This quantity is maximized when we have $|T| = 2$ tiers that contain $\frac{n}{2}$ items each (rounding if $n$ is odd). In this case, the maximum value is $\frac{n}{4}$ (or slightly below if $n$ is odd). The worst case is tight, achieved with two tiers, each with half the items, and an even number of items. $\square$

## C.4 On Stability with Respect to New Items

We start with a simple counterexample to show that selective rankings do not satisfy the "independence of irrelevant alternatives" axiom [4].

**Example C.4** (Selective Rankings do not Satisfy IIA)**.** Consider a preference aggregation task where we have pairwise preferences from 2 users for 2 items $i$ and $j$ where both users agree that $i \succ j$.

$$\text{User 1}: \quad i \succ j$$
$$\text{User 2}: \quad i \succ j$$

In this case, every $\tau$-selective ranking would be $\pi_{i,j}(T) = 1$ for any $\tau \in [0, 0.5)$.

Suppose we elicit preferences for a third item $z$, and discover that each user asserts that $z$ is equivalent to a different item:

$$\text{User 1}: \quad i \sim z \succ j \quad \longleftrightarrow \quad i \succ j \quad z \succ j \quad i \sim z$$
$$\text{User 2}: \quad i \succ j \sim z \quad \longleftrightarrow \quad i \succ j \quad j \sim z \quad i \succ z$$

In this case, every $\tau$-selective ranking would be $\pi_{i,j}(T) = 0$ for all $\tau \in [0, \frac{1}{2})$. This violates IIA because the relative comparison $\pi_{i,j}(T)$ changes depending on the preferences involving $z$.

---

[2]Given a dataset of complete preferences and $\tau \in [0, \frac{1}{2})$, at least one of the following must hold: $\sum_{k\in[p]} \mathbb{I}\left[\pi_{i,j}^k \neq 1\right] > \tau p$ or $\sum_{k\in[p]} \mathbb{I}\left[\pi_{i,j}^k \neq -1\right] > \tau p$. This is because for the former claim to be true, we'd need at least $(1-\tau)p$ preferences to be 1, which then forces the latter claim to be false because we've set $(1-\tau)p > \tau p$ values to be something other than -1.

**Proposition C.5.** *Consider a preference aggregation task where for a given $\tau \in [0, \frac{1}{2})$ we construct a selective ranking $S_n$ using a dataset $\mathcal{D}$ of complete pairwise preferences from $p$ users over $n$ items in the itemset $[n]$. Say we elicit pairwise preferences from all $p$ users with respect to a new item $n + 1$ and construct a selective ranking $S_{n+1}$ for the same $\tau$ over the new itemset $[n + 1]$. Given any two items $i, j \in [n]$, we have that*

$$(\pi_{i,j}(S_{n+1}) = \pi_{i,j}(S_n)) \vee (\pi_{i,j}(S_{n+1}) = 0).$$

*Proof.* It is sufficient to show the following:

- When $\pi_{i,j}(S_n) \neq -1$, we never have $\pi_{i,j}(S_{n+1}) = -1$

- When $\pi_{i,j}(S_n) \neq 1$, we never have $\pi_{i,j}(S_{n+1}) = 1$.

Given a dataset of complete pairwise preferences and $\tau \in [0, \frac{1}{2})$, at least one of the following conditions must hold:

$$\text{Condition I:} \qquad \sum_{k \in [p]} \mathbb{I}\left[\pi_{i,j}^k \neq 1\right] > \tau p$$

$$\text{Condition II:} \qquad \sum_{k \in [p]} \mathbb{I}\left[\pi_{i,j}^k \neq -1\right] > \tau p$$

This is because for Condition I to be False, we would need at least $(1 - \tau)p$ preferences to be 1, which then forces Condition II to be true because we have set $(1 - \tau)p > \tau p$ values to be something other than $-1$.

Consider WLOG that Condition I holds. If $\sum_{k \in [p]} \mathbb{I}\left[\pi_{i,j}^k \neq 1\right] > \tau p$, then we know that $\pi_{i,j}(S_n) \neq 1$. Otherwise we would violate the disagreement constraint in SPA$_\tau$. Note that eliciting preferences for a new item does not change $\sum_{k \in [p]} \mathbb{I}\left[\pi_{i,j}^k \neq 1\right]$. So we still have $\sum_{k \in [p]} \mathbb{I}\left[\pi_{i,j}^k \neq 1\right] > \tau p$, and we still have $\pi_{i,j}(S_{n+1}) \neq 1$. Thus, we have that both $\pi_{i,j}(S_n) \neq 1$ and $\pi_{i,j}(S_{n+1}) \neq 1$. We can apply a symmetric argument to show Condition II holds. In this case, we would have that $\sum_{k \in [p]} \mathbb{I}\left[\pi_{i,j}^k \neq -1\right] > \tau p$ and see that both $\pi_{i,j}(S_n) \neq -1$ and $\pi_{i,j}(S_{n+1}) \neq -1$.

This guarantees that the claim of Proposition 4.3 cannot be violated. When $\pi_{i,j}(S_n) = 0$ so too does $\pi_{i,j}(S_{n+1}) = 0$. When $\pi_{i,j}(S_n) \neq -1$ we never have $\pi_{i,j}(S_{n+1}) = -1$, when $\pi_{i,j}(S_n) \neq 1$ we never have $\pi_{i,j}(S_{n+1}) = 1$. Thus we have proven the claim by cases. □

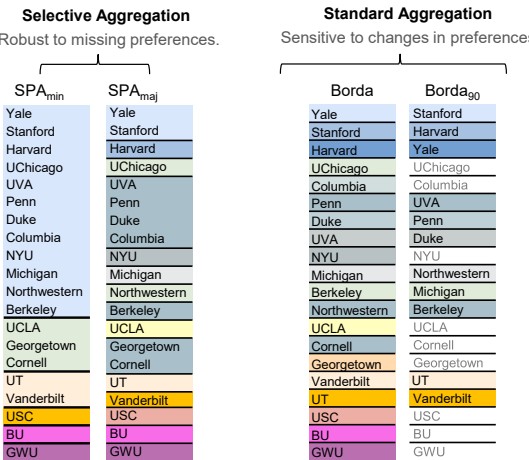

Figure 6: Consensus rankings of U.S. law schools from selective preference aggregation and standard voting rules for the `lawschool` dataset. On the left, we show selective rankings $SPA_{min}$ and $SPA_{maj}$ for dissent values of $\tau_{min} = \frac{1}{5}$ and $\tau_{max} = \frac{2}{5}$. On the right we see Borda on the full dataset, and $Borda_{90}$ after removing 10% of pairwise preferences — illustrating sensitivity to missing data.

## D  Supplementary Material for Sections 5 and 6

In what follows, we include additional details and results for the experiments in Section 5 and our demonstration in Section 6.

### D.1  Descriptions of Datasets

| Dataset | $p$ | $n$ | Format | Description |
|---|---|---|---|---|
| nba | 7 Coaches | 100 Voters | Ballots | 2021 NBA Coach of the Year Award, where sports journalists vote for the top 3 coaches |
| lawschool | 5 Rankings | 26 Schools | Rankings | Top U.S. law schools ranked by 5 organizations based on academic performance, reputation, and other metrics in 2023. |
| survivor | 6 Fans | 39 Seasons | Rankings | Rankings task where 6 fans of the show Survivor rank seasons 1-40 from best to worst. |
| sushi | 5,000 Respondents | 10 Sushi Types | Pairwise | Benchmark recommendation dataset collected in Japan, where participants provided pairwise preferences over 10 different types of sushi: ebi (shrimp), anago (sea eel), maguro (tuna), ika (squid), uni (sea urchin), ikura (salmon roe), tamago (egg), toro (fatty tuna), tekka-maki (tuna roll), and kappa-maki (cucumber roll). |
| csrankings | 5 Subfields | 175 Departments | Rankings | Rankings of computer science departments from csrankings.org based on research output in AI, NLP, Computer Vision, Data Mining, and Web Retrieval. |

Table 4: Overview of datasets. We consider five datasets from salient use cases of preference aggregation.

### D.2  List of Metrics

In what follows, we provide detailed descriptions of the metrics in Table 1.

| Metric | Formula | Description |
|---|---|---|
| AbstentionRate$(T)$ | $\dfrac{1}{n(n-1)}\sum\limits_{i,j\in[n]}\mathbb{I}\left[\pi_{i,j}(T)=\bot\right]$ | Given a selective ranking over $n$ items $T$, the abstention rate represents the fraction of pairwise comparisons where we abstain. |
| DisagreementRate$(T,\mathcal{D})$ | $\dfrac{1}{n(n-1)p}\sum\limits_{k\in[p]}\sum\limits_{i,j\in[n]}\mathbb{I}\left[\pi_{i,j}^k\neq\pi_{i,j}(T),\pi_{i,j}(T)\neq\bot\right]$ | Given a ranking over $n$ items $T$, the *disagreement rate* represents the fraction of individual preferences in $\mathcal{D}$ that disagree with the collective preferences in $T$. |
| #Tiers$(S_\tau)$ | $|S_\tau|$ | Given a selective ranking $S_\tau$, the number of tiers. For standard methods, each rank is converted to a tier. |
| #TopItems$(S_\tau)$ | $|T_1|$ | Given $S_\tau=(T_1,\ldots,T_m)$, the number of items in the top tier. For standard methods, each rank is converted to a tier. |
| DisagreementPerUser$(T,\mathcal{D})$ | $\underset{k\in[p]}{\mathrm{median}}\ \dfrac{1}{n(n-1)/2}\sum\limits_{i,j\in[n]}\mathbb{I}\left[\pi_{i,j}^k\neq\pi_{i,j}(T)\right]$ | The median fraction of preference violations across users. |
| $\Delta$ Sampling $(T,\mathcal{D})$ | $\underset{b\in\{1,\ldots,N_b\}}{\mathrm{median}}\left[\dfrac{\sum_{i,j\in[n]}\mathbb{I}\left[T_{i,j}\neq T_{i,j}^b\wedge T_{i,j}\neq 0\wedge T_{i,j}^b\neq 0\right]}{\sum_{i,j\in[n]}\mathbb{I}\left[T_{i,j}\neq 0\right]}\right]$ | Given the ranking produced on the full dataset $T$, the median proportion of collective preferences that are inverted when we drop 10% of preferences. We construct a bootstrap estimate by applying the method to $N_b$ datasets where we randomly drop 10% of all preferences and obtain $N_b$ rankings $\{T^1,\ldots,T^{N_b}\}$. |
| $\Delta$ Adversarial $(T,\mathcal{D})$ | $\underset{b\in\{1,\ldots,N_b\}}{\max}\left[\dfrac{\sum_{i,j\in[n]}\mathbb{I}\left[T_{i,j}\neq T_{i,j}^b\wedge T_{i,j}\neq 0\wedge T_{i,j}^b\right]\neq 0}{\sum_{i,j\in[n]}\mathbb{I}\left[T_{i,j}\neq 0\right]}\right]$ | Given the original ranking $T$, the *maximum* proportion of collective preferences inverted when we flip 10% of individual preferences. We construct a bootstrap estimate where we first apply the method to $N_b$ datasets where we randomly flip 10% of all preferences and obtain $N_b$ rankings $\{T^1,T^2,\ldots,T^{N_b}\}$. |

Table 5: Metrics used to evaluate comparability, disagreement, and robustness of rankings in Table 1 and Appendix D.4

### D.3 Selective Ranking Paths

We present the solution paths of selective rankings for each dataset in Section 5 in Fig. 7 to Fig. 11.

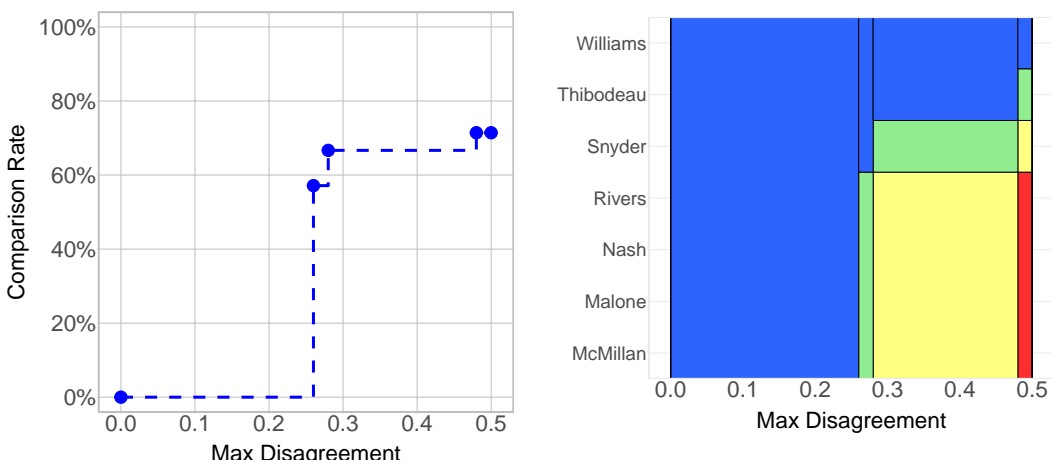

Figure 7: Selective rankings for the `nba` dataset ($n=7$ items and $p=100$ users). We show the tradeoff between comparison and disagreement (left) and the unique rankings over the dissent path (right).

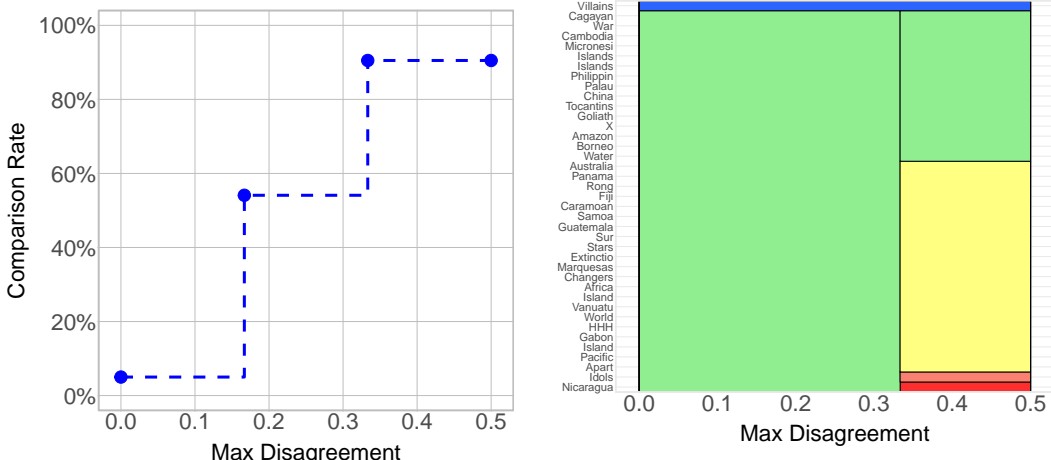

Figure 8: Selective rankings for the `survivor` dataset ($n = 39$ items and $p = 6$ users). We show the tradeoff between comparison and disagreement (left) and the unique rankings over the dissent path (right).

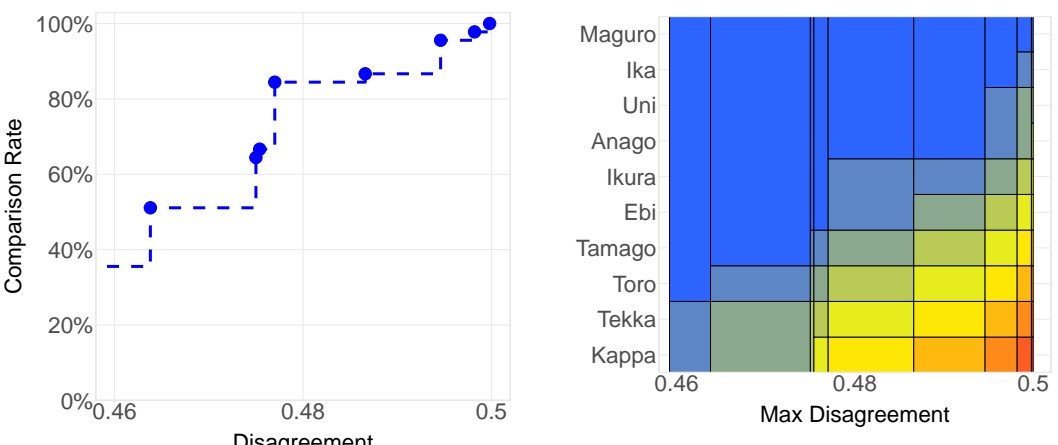

Figure 9: Selective rankings for the `sushi` dataset ($n = 10$ items and $p = 5000$ users). We show the tradeoff between comparison and disagreement (left) and the unique rankings over the dissent path (right). Note that only a subset of dissent values are shown for clarity, focusing on the largest areas of change.

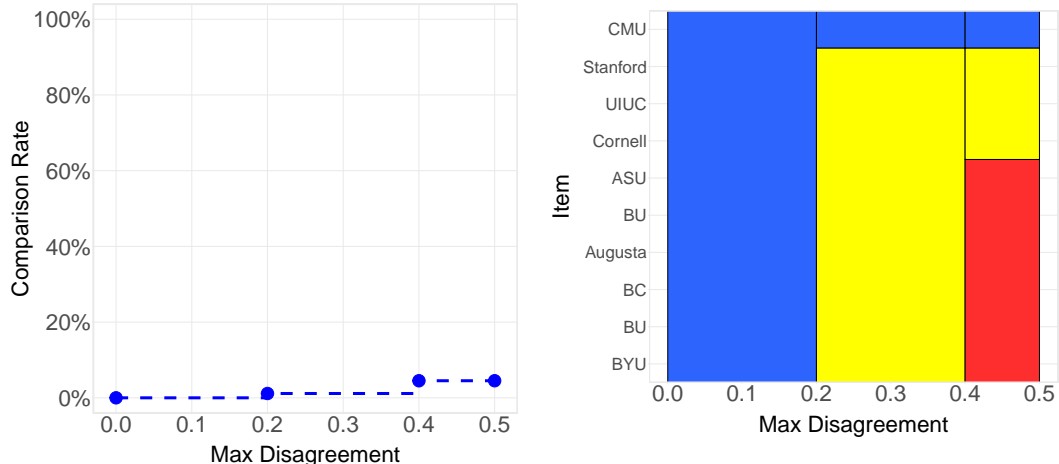

Figure 10: Selective rankings for the `csrankings` dataset ($n = 175$ items and $p = 5$ users). We show the tradeoff between comparison and disagreement (left) and the unique rankings over the dissent path (right). We show the top 10 items, sorted by tier and alphabetically within each tier.

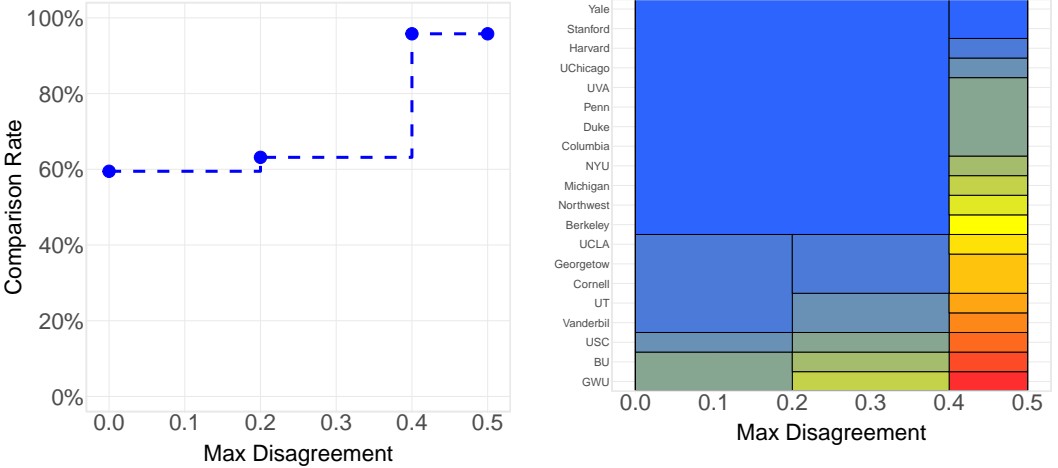

Figure 11: Selective rankings for the `lawschool` dataset ($n = 20$ items and $p = 5$ users). We show the tradeoff between comparison and disagreement (left) and the unique rankings over the dissent path (right).

## D.4 Expanded Table of Results

We include an expanded version of our results for all methods and all datasets in Appendix D.4. This table covers the same results as in Table 1, but includes the following additional metrics:

1. $\Delta$ *Abstentions [Intervention]*, which measures the proportion of strict collective preferences (e.g., $A \succ B$ or $A \prec B$) that turn into ties or abstentions in the ranking that we obtain after running the method on a modified dataset.

2. $\Delta$ *Specifications [Intervention]*, which measures the proportion of ties or abstentions that turn into ties or abstentions in the ranking that we obtain after running the method on a modified dataset.

We report these values for same interventions we consider in Section 5, namely: *Sampling*, where we run the method on a dataset where we randomly omit 10% of individual preferences; and *Adversarial*, where we run the method on a dataset where we randomly flip 10% of individual preferences. Each value corresponds to a bootstrap estimates where we perform the same estimate 100 times. For clarity, we list the $\Delta$ − Sampling as $\Delta$ − Inversions − −Sampling, and $\Delta$ − Adversarial − −Inversions.

| | | Selective | | | Standard | | | | |
|---|---|---|---|---|---|---|---|---|---|
| **Dataset** | **Metrics** | SPA$_0$ | SPA$_{min}$ | SPA$_{maj}$ | Borda | Copeland | MC4 | KemenyExact | KemenyHeuristic |
| nba
$n = 7$ items
$p = 100$ users
28.6% missing
NBA [56] | Disagreement Rate | 0.0% | 2.0% | 6.4% | 8.3% | 8.3% | 7.9% | 8.1% | 8.1% |
| | Median Disagreement per User | 0.0% | 0.0% | 4.8% | 4.8% | 4.8% | 9.5% | 9.5% | 9.5% |
| | Abstention Rate | 100.0% | 42.9% | 28.6% | – | – | – | – | – |
| | # Tiers | 1 | 2 | 4 | 7 | 7 | 6 | 7 | 7 |
| | # Top Items | 7 | 3 | 1 | 1 | 1 | 1 | 1 | 1 |
| | Dissent | 0.0000 | 0.2600 | 0.4900 | – | – | – | – | – |
| | Δ Inversions Sampling | 0.0% | 0.0% | 0.0% | 4.8% | 4.8% | 0.0% | 4.8% | 4.8% |
| | Δ Inversions Adversarial | 0.0% | 0.0% | 0.0% | 19.0% | 19.0% | 19.0% | 14.3% | 14.3% |
| | Δ Specifications Sampling | 0.0% | 9.5% | 0.0% | 0.0% | 0.0% | 4.8% | 0.0% | 0.0% |
| | Δ Specifications Adversarial | 0.0% | 9.5% | 0.0% | 0.0% | 0.0% | 4.8% | 0.0% | 0.0% |
| | Δ Abstentions Sampling | 0.0% | 0.0% | 28.6% | 0.0% | 0.0% | 0.0% | 0.0% | 0.0% |
| | Δ Abstentions Adversarial | 0.0% | 19.0% | 28.6% | 0.0% | 4.8% | 33.3% | 0.0% | 0.0% |
| survivor
$n = 39$ items
$p = 6$ users
0.0% missing
Purple Rock [57] | Disagreement Rate | 0.0% | 0.2% | 0.2% | 6.8% | 6.6% | 6.4% | 6.7% | 6.7% |
| | Median Disagreement per User | 0.0% | 0.1% | 0.1% | 7.2% | 7.1% | 6.8% | 7.1% | 7.1% |
| | Abstention Rate | 94.9% | 42.5% | 42.5% | – | – | – | – | – |
| | # Tiers | 2 | 5 | 5 | 39 | 36 | 35 | 39 | 39 |
| | # Top Items | 1 | 1 | 1 | 1 | 1 | 1 | 1 | 1 |
| | Dissent | 0.0000 | 0.1667 | 0.3333 | – | – | – | – | – |
| | Δ Inversions Sampling | 0.0% | 0.0% | 0.0% | 1.3% | 0.8% | 0.8% | 0.9% | 0.9% |
| | Δ Inversions Adversarial | 0.0% | 0.0% | 0.0% | 2.6% | 1.8% | 3.1% | 1.6% | 1.6% |
| | Δ Specifications Sampling | 0.0% | 0.0% | 0.0% | 0.0% | 0.4% | 0.1% | 0.0% | 0.0% |
| | Δ Specifications Adversarial | 0.0% | 5.1% | 0.0% | 0.0% | 0.4% | 0.3% | 0.0% | 0.0% |
| | Δ Abstentions Sampling | 0.0% | 52.4% | 57.5% | 0.0% | 0.1% | 80.0% | 0.0% | 0.0% |
| | Δ Abstentions Adversarial | 0.0% | 57.5% | 57.5% | 0.0% | 0.4% | 89.5% | 0.4% | 0.4% |
| lawschool
$n = 20$ items
$p = 5$ users
0% missing
LSData [58] | Disagreement Rate | 0.0% | 0.3% | 3.1% | 4.7% | 4.2% | 4.2% | 4.1% | 4.1% |
| | Median Disagreement per User | 0.0% | 0.0% | 1.6% | 4.2% | 2.6% | 2.6% | 2.1% | 2.1% |
| | Abstention Rate | 40.5% | 36.8% | 4.2% | – | – | – | – | – |
| | # Tiers | 4 | 6 | 15 | 20 | 20 | 19 | 20 | 20 |
| | # Top Items | 12 | 12 | 2 | 1 | 1 | 1 | 1 | 1 |
| | Dissent | 0.0000 | 0.2000 | 0.4000 | – | – | – | – | – |
| | Δ Inversions Sampling | 0.0% | 0.0% | 0.0% | 1.6% | 1.1% | 0.5% | 29.5% | 29.5% |
| | Δ Inversions Adversarial | 0.0% | 0.0% | 0.0% | 3.7% | 2.6% | 2.6% | 45.8% | 45.8% |
| | Δ Specifications Sampling | 0.0% | 11.1% | 0.0% | 0.0% | 0.0% | 0.0% | 0.0% | 0.0% |
| | Δ Specifications Adversarial | 0.0% | 0.0% | 0.5% | 0.0% | 0.0% | 0.0% | 0.0% | 0.0% |
| | Δ Abstentions Sampling | 59.5% | 28.2% | 95.8% | 0.0% | 0.0% | 55.8% | 0.0% | 0.0% |
| | Δ Abstentions Adversarial | 59.5% | 0.0% | 95.8% | 0.0% | 1.6% | 64.2% | 0.0% | 0.0% |
| csrankings
$n = 175$ items
$p = 5$ users
0.0% missing
Berger [59] | Disagreement Rate | 0.0% | 0.0% | 0.1% | 12.3% | 12.2% | 12.2% | – | 13.7% |
| | Median Disagreement per User | 0.0% | 0.0% | 0.1% | 12.3% | 12.6% | 12.3% | – | 13.5% |
| | Abstention Rate | 100.0% | 98.9% | 95.5% | – | – | – | – | – |
| | # Tiers | 1 | 2 | 3 | 175 | 168 | 170 | – | 175 |
| | # Top Items | 175 | 1 | 1 | 1 | 1 | 1 | – | 1 |
| | Dissent | 0.0000 | 0.2000 | 0.4000 | – | – | – | – | – |
| | Δ Inversions Sampling | 0.0% | 0.0% | 0.0% | 0.8% | 0.8% | 0.1% | – | 9.0% |
| | Δ Inversions Adversarial | 0.0% | 0.0% | 0.0% | 3.1% | 1.7% | 0.1% | – | 11.1% |
| | Δ Specifications Sampling | 0.0% | 0.0% | 0.0% | 0.0% | 0.1% | 94.4% | – | 0.0% |
| | Δ Specifications Adversarial | 0.0% | 0.0% | 0.0% | 0.0% | 0.1% | 94.4% | – | 0.0% |
| | Δ Abstentions Sampling | 0.0% | 1.1% | 4.5% | 0.0% | 0.0% | 0.0% | – | 0.0% |
| | Δ Abstentions Adversarial | 0.0% | 0.0% | 4.5% | 0.0% | 0.1% | 0.0% | – | 0.0% |
| sushi
$n = 10$ items
$p = 5,000$ users
0.0% missing
Kamishima [60] | Disagreement Rate | 0.0% | 13.6% | 42.6% | 42.6% | 42.6% | 42.6% | 42.6% | 42.6% |
| | Median Disagreement per User | 0.0% | 13.3% | 42.2% | 42.2% | 42.2% | 42.2% | 42.2% | 42.2% |
| | Abstention Rate | 100.0% | 64.4% | 0.0% | – | – | – | – | – |
| | # Tiers | 1 | 2 | 10 | 10 | 10 | 10 | 10 | 10 |
| | # Top Items | 10 | 8 | 1 | 1 | 1 | 1 | 1 | 1 |
| | Dissent | 0.0000 | 0.0020 | 0.4998 | – | – | – | – | – |
| | Δ Inversions Sampling | 0.0% | 0.0% | 0.0% | 0.0% | 0.0% | 2.2% | 2.2% | 2.2% |
| | Δ Inversions Adversarial | 0.0% | 0.0% | 0.0% | 2.2% | 2.2% | 11.1% | 11.1% | 11.1% |
| | Δ Specifications Sampling | 0.0% | 0.0% | 0.0% | 0.0% | 0.0% | 0.0% | 0.0% | 0.0% |
| | Δ Specifications Adversarial | 0.0% | 0.0% | 0.0% | 0.0% | 0.0% | 0.0% | 0.0% | 0.0% |
| | Δ Abstentions Sampling | 0.0% | 35.6% | 100.0% | 0.0% | 0.0% | 0.0% | 0.0% | 0.0% |
| | Δ Abstentions Adversarial | 0.0% | 0.0% | 100.0% | 0.0% | 0.0% | 15.6% | 0.0% | 0.0% |

## D.5 Supplementary Material for Section 6

**Selective Aggregation with Binary Annotations**    A key challenge in applying SPA to the DICES dataset is that it elicits categorical labels for each item individually, rather than comparative ratings. This conversion can create unnecessary equivalence, where a pairwise preference is inferred as a tie ($\pi_{i,j}^k = 0$). This is not a reflection of a user's true judgment but an artifact of two limitations: (1) users annotate items individually rather than comparing them, and (2) the annotations are restricted to $\{0,1\}$ instead of granular ratings. For example, a user may believe item A is significantly more toxic than item B, but the conversion results in a tie if both were labeled "toxic" a distinction that is lost in this setting.

We address this by running a variant of selective aggregation where we construct aggregate labels from users who express a strict preference between items $- i \succ j$ or $j \succ i$. In addition, we assume

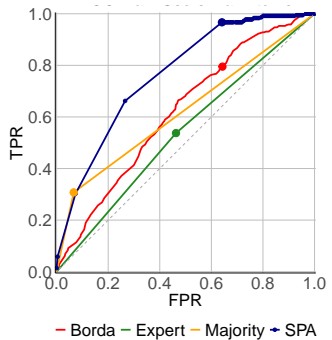

Figure 12: ROC model curves on the training set for all four methods. We highlight the label for each method closest to tpr> $90\%$ on labels with a large dot. $f^{\text{SPA}}$ is the only method whose chosen operating point keeps the true-positive rate above 80 % on the model output while controlling FPR.

that users who have not asserted an opinion (because of dataset scope) are "deferring judgment" to those who have.

For each pair of items $i, j \in [n]$, we define:

- $s_{i,j} := \sum_{k \in [p]} \mathbb{I}\left[\pi_{i,j}^k = 1\right]$ denote number of users who strictly prefer item $i$ to item $j$

- $s_{j,i} := \sum_{k \in [p]} \mathbb{I}\left[\pi_{i,j}^k = -1\right]$ denote the number of users who strictly prefer item $j$ to item $i$.

- The aggregate preference weight $w_{i,j}$ as the proportion of users who strictly prefer $i$ to $j$ among those who expressed a strict preference, scaled to $n$ items. Note that all item pairs had at least 1 preference:

$$w_{i,j} := n \cdot \frac{s_{i,j}}{s_{i,j} + s_{j,i}}$$

In this setup, the dissent parameter $\tau$ no longer maintains its standard interpretation because users may not assign a preference to each item, and items may be assigned different weights. As a result, we produce selective rankings for all possible dissent parameters that lead to a connected graph in Algorithm 2. In this case, the maximum dissent value is set to a threshold value where Line 4 returns a disconnected graph.

## D.6 Model Training

All experiments used 5-fold cross-validation on the training split. We fine-tuned a BERT-Mini model; all fine-tuning experiments used 5-fold cross-validation on the training split. We optimized with a learning rate of $2 \times 10^{-5}$ for up to 25 epochs, employing early stopping. We trained in mini-batches of size 16 and enabled oversampling of minority classes in each batch.

