# OpenReview forum: "Selective Preference Aggregation"
_NeurIPS.cc/2025/Workshop/Reliable_ML — NeurIPS 2025 - Reliable ML Workshop_

### Official Review · Reviewer_BWGx · 2025-09-20
**Interesting idea about elective ranking**

**Rating:** 7
**Confidence:** 3

**Review:**

Pros

Proposes selective rankings on pairs with too much dissent, making disagreement explicit rather than hiding it in a forced total order.

Simple, fast algorithm: constructs a directed graph from preference counts, collapses strongly connected components into tiers, then topologically sorts tiers, transparent and easy to implement.


Broad applicability on alignment, labeling, rankings where disagreement are inherent and provides an open-source library.

Cons

Lacks novelty partial orders and tiered structures exist, novelty is in problem framing

---

### Official Review · Reviewer_Euy3 · 2025-09-24
**This paper introduces Selective Preference Aggregation (SPA), a framework for aggregating human preferences into partial orders rather than total rankings.**

**Rating:** 7
**Confidence:** 3

**Review:**

Strengths

- The paper is in general well-written, makes it easy for the readers.
- Algorithm 1 is also very intuitive. The time complexity was not discussed, however, shouldn't be hard to compute.
- Provides both algorithmic solutions and formal guarantees (e.g., stability with missing data, recovery of Condorcet winners).
- Demonstrates SPA on diverse datasets and shows its impact in LLM alignment tasks. Also the code has been open-sourced to enable further research.

Questions

- While baselines like Borda and Kemeny are included, there may be other algorithms that may be more competitive that do not necessarily propose algorithms for this problem, but for an approximation of total ordering that could achieve good partial orderings?